# Peer Effects on Farmers’ Purchases of Policy-Based Planting Farming Agricultural Insurance: Evidence from Sichuan Province, China

**DOI:** 10.3390/ijerph19127411

**Published:** 2022-06-16

**Authors:** Xueling Bao, Fengwan Zhang, Shili Guo, Xin Deng, Jiahao Song, Dingde Xu

**Affiliations:** 1College of Management, Sichuan Agricultural University, Chengdu 611130, China; 2020209052@stu.sicau.edu.cn (X.B.); 2020309005@stu.sicau.edu.cn (F.Z.); 2School of Economics, Southwestern University of Finance and Economics, Chengdu 610074, China; guoshili@swufe.edu.cn; 3College of Economics, Sichuan Agricultural University, Chengdu 611130, China; dengxin@sicau.edu.cn; 4Sichuan Center for Rural Development Research, College of Management, Sichuan Agricultural University, Chengdu 611130, China

**Keywords:** policy-based planting agricultural insurance, peer effect, mechanism analysis, Sichuan, China

## Abstract

The tendency to conform with peers, and learning by imitation, have become new influencing factors that affect farmers’ purchases of policy-based planting agricultural insurance. Based on the survey data of 540 farmers in Sichuan Province in 2021, this study empirically analyzed the impact of peer effects on farmers’ purchases of policy-based planting agricultural insurance and its mechanism. The results show that: (1) Regardless of whether farmers’ relatives and friends visit during the New Year period, the purchase of policy-based planting agricultural insurance by relatives and friends will positively and significantly affect the purchasing behavior of the farmers. (2) The impact of the peer effect on the behavior of farmers purchasing policy-based planting agricultural insurance is related to the relationship between the strengths and weaknesses. (3) The results of the mechanism analysis show that, through the mediating variables of social network and trust, the influence of the peer effect is weakened. (4) Heterogeneity analysis shows that farmers having a larger land scale and higher educational background are more influenced by the same peer effect. The results of the study emphasize the importance of the peer effect on the behavior of farmers purchasing policy-based planting agricultural insurance, and can provide a decision-making reference for the formulation of related policies.

## 1. Introduction

China is a large agricultural country, and its agricultural development is extremely dependent on natural conditions such as temperature, precipitation, and terrain. Furthermore, China has a vast territory and complex terrain. Various meteorological and natural disasters frequently occur, which are spread over a wide range and cause huge losses. Therefore, the development of China’s agriculture has faced various difficulties and challenges [1,2,3]. The risks of agricultural production and climate disasters caused by climate change lead to damage to agricultural production. According to Chinese statistics, in 2019, the crop-producing areas affected by droughts and floods amounted to 3332.3 and 2611.8 thousand hectares, respectively [4]. In order to reduce the agricultural losses incurred by farmers caused by various meteorological disasters and natural disasters, many countries have implemented and promoted agricultural insurance as a method to protect farmers’ agricultural production [5]. In China, in order to stabilize the income of farmers growing grain, support the development of modern agriculture, and ensure food security, since 2007, the departments of finance, agriculture, and rural areas have implemented the relevant arrangements of the Party Central Committee and the State Council, and implemented the planting agricultural insurance premium subsidy. This has been undertaken in accordance with the principles of “government guidance, market operation, independent and voluntary, and coordinated advancement”. This scheme provides certain premium subsidies for insured farmers; for example, at present, for policy-based planting farming agricultural insurance, the central financial subsidy is 40%, the provincial financial subsidy is 25%, and the city and county financial subsidy is 15%, and the farmer bears 20%. The scheme also guides and supports farmers to participate in policy-based planting agricultural insurance, and has gradually increased support [6,7]. According to the latest statistics from the Ministry of Agriculture and Rural Affairs, the central government arranged policy-based planting agricultural insurance premium subsidies having a value of RMB 33.345 billion in 2021, an increase of 16.8% over 2020 [8]. It can be seen from the policy layout and capital investment that policy-based planting agricultural insurance, as one of the important risk guarantees for agricultural and rural development, is of great significance to China’s implementation of its rural revitalization strategy.

However, since the implementation and promotion of policy-based planting agricultural insurance, only a single type of policy-based planting agricultural insurance has been offered, leading to a shortage in supply. The insurance has also been inconsistent with the actual situation in various places. Furthermore, demand for the insurance has been deficient due to the lack of enthusiasm among farmers to participate in the scheme. As a result of these factors, China’s policy-based planting agricultural insurance remains in a “Double supply and demand” situation [9]. In order to solve the problem of “supply” of policy-based planting agricultural insurance, the number of subsidized varieties of this insurance in China has been expanded from five in the initial planting industry, to 16 in three categories of planting, breeding, and forestry. At the same time, China has also encouraged the development of agricultural product insurance with local advantages and characteristics, and places where conditions permit can provide certain premium subsidies. The scope of compensation for policy-based agricultural insurance includes the following options: The amount of planting insurance is based on the direct physical and chemical costs incurred during the growth period of the insurance subject (including costs of seeds, fertilizers, pesticides, irrigation, machine farming, and mulch film); and the insurance amount for aquaculture is the insured individual physiological value (including purchase and feeding costs). At present, the specific compensation standard of China’s policy-based planting farming insurance is as follows: The starting point for claims is 30%; that is, when the loss rate of the insured crops due to natural disasters ranges from 30% to 70%, the insurance amount and the loss rate are divided by the crop growth period to calculate the compensation. The specific calculation formula for the compensation is: compensation amount = insurance amount in each growth period × loss rate × damaged area. When the loss rate reaches more than 70%, the full amount of insurance is paid according to the crop growth period. Taking rice cultivation as an example, the insurer is not responsible for paying compensation due to natural disasters within the scope of insurance coverage when the loss rate is less than 30%; if the actual loss rate is from 30% to 70%, the compensation will be paid proportionally; and if the actual loss rate is 70% or above, full compensation will be paid. When the area of the insured rice plot of each insured farmer is smaller than the actual planting area, the compensation is calculated according to the proportion of the insured area to the actual planting area. To summarize, according to the risk aversion theory, in the context of the threat of natural disasters, and government policies and financial subsidies for premiums, the participation rate of agricultural insurance can theoretically reach 100%. China’s policy-based planting agricultural insurance basically covers grain crops, cash crops, and livestock involved in the planting and breeding industries. However, according to statistics, nearly 33% of farmers in China have not purchased policy-based planting agricultural insurance, and more than 55% of crops are not covered by policy-based planting agricultural insurance [10]. As a result of this anomalous phenomenon, which is contrary to reality, scholars have closely examined why, under the threat of agricultural disasters, and such a generous national policy guarantee, some farmers are reluctant to actively participate in insurance.

In the existing research, most scholars try to conduct front-end analysis of the factors affecting the purchase of policy-based planting agricultural insurance from the perspectives of farmers’ perception of meteorological disasters, local government financial expenditures, personal characteristics, and family profiles. For example, Zheng et al. [11] found that climate risk induces loss aversion among farmers, which translates into demand for policy-based planting agricultural insurance. Yue and Liu [9] found that local government fiscal expenditures and policy-based planting agricultural insurance purchases are positively and significantly correlated. Zhang and Chen [12] found that farmers’ income levels and farmland management scales have a positive and significant correlation with their understanding of policy-based planting agricultural insurance and their willingness to purchase insurance. Other scholars have tried to conduct back-end research on whether farmers’ income, production efficiency, investment, and land changed after purchasing policy-based planting agricultural insurance. Through a case study in Japan, Yamauchi and Toyoji [13] empirically found that the purchase of policy-based planting agricultural insurance can further stabilize farmers’ income and reduce farmers’ disaster risk losses. Enjolras et al. [14] found that farmers in Italy and France can effectively reduce income volatility and stabilize agricultural income by purchasing crop insurance. Liu et al. [15] found that, regarding farmers’ purchases of policy-based planting agricultural insurance, the breadth and depth of coverage has a significant intermediary effect on the net operating income of farmers, and the current low security of policy-based planting agricultural insurance in China has inhibited the increase in the net operating income of farmers. Ren et al. [16] found that the impact on the productivity of farmers of the security level of policy-based planting agricultural insurance has an inverted U-shape.

In fact, farmers’ purchases of policy-based planting agricultural insurance are determined by a purchase behavior and a decision-making behavior. Among the factors that affect individual participation in decision making, the peer effect is important. In theory, the peer effect specifically refers to the fact that people around the micro-individual affect the decision-making behavior of the micro-subject through the two mechanisms of information transmission and social norms [17,18,19,20]. Studies have shown that peer effects play an important role in micro-decisions, such as investment and risk aversion [21,22,23,24,25]. The peer effect also leads to the formation of behavioral norms within the group, which in turn affects the decision-making behavior of each individual in that group [26]. Previous scholars have also conducted similar research on the impact of peer effects on farmers’ purchasing decisions. For example, Zhang and Zhu [27] found that there is a significant positive correlation between the insurance participation behavior of residents in the same village and the individual insurance participation behavior. The transmission of information among residents in the same village and the social norms formed within them are important channels in which the peer effect can play a role.

Due to various reasons, such as the manner and intensity of policy propaganda in various parts of China, and the enthusiasm of farmers to actively study, farmers often lack a correct and comprehensive understanding of policy insurance such as policy-based planting agricultural insurance. Therefore, when farmers decide whether to participate in insurance, they often need to obtain information from the decision-making behavior of surrounding people. However, from the perspective of peer effects, few rigorous studies have been undertaken on policy-based planting agricultural insurance. Therefore, this study first examined whether the peer effect has an impact on farmers’ purchases of policy-based planting agricultural insurance and farmers’ decision making. Secondly, the theoretical marginal contribution of this study is the distinction of the peer effect; this is in contrast to the previous research, which often distinguishes farmers based on economic strength or location factors. Furthermore, the current study paid attention to the “circle effect” within the peer effect. Additionally, based on the use of binary logistic regression to specifically explore the relationship between the three core variables that reflect the peer effect and farmers’ purchases of policy-based planting agricultural insurance, this study also used the propensity score matching method (PSM model) to deal with the estimation error caused by sample selection bias. The aim of this was to better present the average processing impact of the peer effect on the purchase of policy-based planting agricultural insurance by the interviewed farmers. Finally, in order to ensure the rigor and integrity of the empirical results, sample-based heterogeneity analysis and mechanism analysis were also used in this study, and the heterogeneity and specific mechanism of the peer effect in farmers’ insurance purchase decisions were analyzed.

## 2. Materials and Methods

### 2.1. Data Sources

The data used in this study were taken from a survey conducted by the research group in Sichuan Province in July 2021. The survey method used was one-to-one, face-to-face interviews. The survey content involved farmers’ livelihood capital, insurance purchase willingness and behavior, etc. In order to ensure the representativeness of the survey samples, the author’s research group adopted the method of stratified equal probability random sampling to determine the survey samples. The specific sampling process was as follows:

First, according to the two indicators of topography and the value of per capita industrial output, the 183 districts and counties in Sichuan Province were divided into three groups by the method of cluster analysis. A district and county was then randomly selected from each group. Finally, three sample districts and counties were randomly selected from the groups, namely, Jiajiang County, Yuechi County, and Gaoxian County, which represent the plain counties, hilly counties, and mountainous counties, respectively, in terms of the topography. Next, all the townships in each sample area and county (41 towns and 2 blocks in total) were randomly divided into three groups by the same method, one of which was randomly selected from each group to obtain three townships; finally, nine sample towns were obtained. After the sample townships were determined, the villages in each sample township were divided into three categories: good, medium, and poor, according to the differences in the economic development level of the villages within the township, and the distance from the township government center. One village was then randomly selected from each category as the sample villages, to obtain 27 villages. After the sample villages were determined, before the formal investigation, the team members of the front station were arranged to conduct the pre-investigation. The roster of farmers in the sample villages was obtained from the village cadres, and used for the random selection from each sample village according to a preset random number table. Thus, 20 households were selected as sample households. Finally, 16 researchers who had undergone rigorous training conducted one-on-one, face-to-face research at farmers’ homes. If the investigator was rejected by the farmer after introducing the purpose and significance of the investigation, we selected a new farmer for investigation according to the principle of random sampling. Finally, a sample size of 20 households in each village was obtained. According to the above process, a total of 540 valid farmer household questionnaires were obtained in 3 districts and counties, 9 townships, and 27 villages. The distribution map of the sample townships is shown in Figure 1.

### 2.2. Theoretical Analysis and Research Assumptions

The peer effect, also known as the herd effect [28], refers to the fact that people around the micro-individual affect the decision-making behavior of the micro-subject through the two mechanisms of information transmission and social norms [17,18,19,20]. In a similar environment, the behavior of surrounding people will affect the behavior of the individual to a varying degree, which is manifested as the individual choosing to learn from or imitate others. In addition, the peer effect will also lead to the formation of behavioral norms within the group, which in turn affects the decision-making behavior of each individual in that group [26]. In today’s society, individuals are prone to information asymmetry, due to differences in information media. As a result, individuals have access to limited information, especially in terms of policy knowledge. Farmers, therefore, often lack a correct and comprehensive understanding of policies due to various reasons, such as their enthusiasm for active learning, and may not even know about such policies. In rural areas where the education level is generally low and information is blocked, the peer effect is more obvious because the judgment of individual farmers is often unclear. At the same time, China is traditionally a society of acquaintances. In this environment, individual farmers will divide themselves and outsiders into groups on the basis of mutual understanding [29]. Individual farmers will have more frequent exchanges and interactions with “people around them” who have blood ties or friends. In addition, it is easier for individual farmers to refer to these social groups to make similar behavioral decisions. Therefore, in this study, relatives and friends within the scope of farmers’ understanding were used as the reference for farmers’ individual decision-making behavior, and the peer effect that impacts farmers’ behavior regarding purchases of policy-based planting agricultural insurance was explored.

In fact, varying distances also exist between relatives and friends; that is, strong and weak relationships [30]. From the existing academic research, it can be seen that the strong–weak relationship theory is also applicable to extensive studies on private lending, labor market networks, and agricultural technology diffusion and peer effects [30,31,32,33,34,35,36]. Compared with groups having weak ties among relatives and friends, farmers are usually represented by groups having strong ties, and usually have a higher frequency of interaction and similar attitudes, which can usually be reflected in their daily interactions. In the context of Chinese culture, the most typical communication that can measure intimacy is the visit during the New Year period [37,38]. Based on this, this study assumed that relatives and friends who visit during the New Year period may have a greater impact on farmers’ purchases of policy-based planting agricultural insurance than those who do not.

According to the above theoretical analysis, this study proposed the following research hypotheses.

**Hypothesis** **1** **(H1).**
*The purchase of policy-based planting agricultural insurance by relatives and friends of farmers will positively affect the purchase of policy-based planting agricultural insurance by farmers.*


**Hypothesis** **2** **(H2).**
*The purchase of policy-based planting agricultural insurance by relatives and friends who visit during the New Year period will positively affect farmers’ purchase of policy-based planting agricultural insurance.*


**Hypothesis** **3** **(H3).**
*The purchase of policy-based planting agricultural insurance by relatives and friends who do not visit during the New Year period will positively affect the purchase of policy-based planting agricultural insurance by farmers.*


**Hypothesis** **4** **(H4).**
*The influence of the peer effect on the behavior of farmers purchasing policy-based planting agricultural insurance is related to the strength of the relationship; that is, relatives and friends > relatives and friends who visit during the New Year period > relatives and friends who do not visit during the New Year period.*


### 2.3. Variable Definitions

#### 2.3.1. Core Variable

This study aimed to reveal the influence of the peer effect on farmers’ purchasing behavior regarding policy-based planting agricultural insurance. Therefore, the basis of the research was the question, “Have you purchased policy-based planting agricultural insurance?”. Values were assigned according to the answer results; answering “yes” was assigned a value of 1, otherwise, a value of 0 was assigned. According to the data, 335 of the 540 interviewed farmers purchased policy-based planting agricultural insurance, accounting for about 62%. The core independent variable examined in this study was the peer effect on the purchase of policy-based planting agricultural insurance by farmers. Considering the influence of strong social relations and weak social relations on the purchase of policy-based planting agricultural insurance by farmers, there may be a “circle effect” (refer to He [37], Xiao [38], and other studies). In this study, we measured the peer effect on farmers’ decisions to purchase policy-based planting agricultural insurance through the following indicators: “Have your relatives and friends purchased this type of insurance?", "Have your relatives and friends who have visited New Year’s have purchased this type of insurance?" and "Have your relatives and friends who have not visited during the New Year have purchased this type of insurance?”. Hereinafter, these indicators are referred to as relatives and friends, who visited during the New Year period and who did not visit during the New Year period, respectively, as measured by the decisions of the respondents’ relatives and friends to purchase policy-based planting agricultural insurance.

#### 2.3.2. Control Variables

In order to avoid the influence of other factors that may affect farmers’ decisions to purchase policy-based planting agricultural insurance on the model estimation results, some variables were included in the study as control variables. The specific selection basis was as follows [12,39]: respondents’ personal characteristics (such as gender, age, education level, farming time) and family characteristics (such as operating land area, annual cash income) are often considered to be related to farmers’ agriculture. Insurance purchase behavior is significantly correlated to this, so this study also included these variables as control variables. Furthermore, farmers’ decisions to purchase policy-based planting agricultural insurance are also affected by their personal perceptions and risk aversion [40,41,42,43,44]. Compensation for crop disasters is generally the direct reason for farmers’ purchase of policy-based planting agricultural insurance; thus, farmers’ perceptions of disasters were added to the model as a control variable. In addition, because individuals have different environmental perceptions, and individuals with higher environmental perception levels have a better understanding of agricultural disasters, “you are very worried about the serious impact of climate change on agricultural production” was added to the model as a control variable for environmental perceptions. In addition, the greater the willingness of farmers to continue farming, the more likely they are to buy policy-based planting agricultural insurance. Therefore, the farmers’ willingness to develop rural areas was added to the model as a control variable. For the decision making of the whole family, the inputs and outputs are unknown, and risk-averse decision makers may be more inclined to buy policy-based planting agricultural insurance. In order to avoid the interference caused by the risk tendency, the risk tendency of the respondents was added to the model. In addition, considering that specific regional measures may affect the purchase of policy-based planting agricultural insurance, this study controlled for the regional effect by setting dummy variables of districts and counties to reduce the regression analysis error caused by the inconsistency of regional factors. The variable definitions and descriptive statistical analysis are shown in Table 1.

### 2.4. Research Methods and Models

#### 2.4.1. Binary Logistic Regression

Reviewing the existing literature, under the premise of maximizing utility, discrete choice models are often used to analyze various behaviors of farmers. Specifically, when the dependent variable is binary, that is, farmers are willing or unwilling to buy policy-based planting agricultural insurance, the binary logistic or binary probit model is usually used [12,45,46].

Therefore, this study adopted a binary logistic model to estimate the influence of the peer effect (including conformity to relatives, neighbors, wealthy villagers, and village cadres) on farmers’ decisions to purchase policy-based planting agricultural insurance. In this model, the behavior of farmers purchasing policy-based planting agricultural insurance is the dependent variable y, and the three core explanatory variables that reflect the cohort effect are $x_{1},x_{2}$, and $x_{3}$.
(1)$$\left\{\begin{array}{c} y_{i}=1\;\;\;\;The\;interviewed\;farmers\;purchased\;agricultural\;insurance\;\;\;\;\;\;\;\;\;\;\;\;\\ y_{i}=0\;\;\;\;The\;interviewed\;farmers\;did\;not\;purchase\;agricultural\;insurance \end{array}\right.$$

Then, the probability P of farmers purchasing policy-based planting agricultural insurance estimated by the binary logistic regression model can be expressed as:(2)$$P(y_{i}=1)=\varphi \left(\alpha +\beta_{1}x_{1}+\beta_{2}x_{2}+\beta_{3}x_{3}+D+\mu \right)$$
where i represents the ith interviewed farmer; $P(y_{i}=1)$ represents the probability of the interviewed farmer purchasing policy-based planting agricultural insurance; α  is the regression intercept; $x_{1},\;x_{2}$, and $x_{3}$ represent three core explanatory variables, respectively: relatives and friends; relatives and friends who visited during the New Year period; and relatives and friends who did not visit during the New Year period. D is the vector of control variables and μ  is the error term.

#### 2.4.2. Propensity Score Matching Method (PSM Model)

The binary logistic regression model can be used to estimate the influence of the effect of peers, namely, relatives and friends, relatives and friends who visited during the New Year period, and relatives and friends who did not visit during the New Year period, on farmers’ purchases of policy-based planting agricultural insurance. In reality, however, there are many similarities and differences between relatives and friends who meet the understanding of farmers, relatives and friends who visit who visited during the New Year period, and relatives and friends who did not visit during the New Year period. This means that the tendency of individual farmers to conform is probably the result of self-selection. In this case, the direct regression results of the binary logistic regression model may suffer from selection bias [47,48]. In order to avoid the estimation error caused by sample selection bias, this study adopted the propensity score matching method developed by Rosenbaum and Rubin [49] to estimate the average treatment effect of the impact of peers on the purchase of policy-based planting agricultural insurance by the interviewed farmers. The specific steps are as follows: In the first step, the binary logistic regression model was used to obtain the probability of the influence of relatives and friends, relatives and friends who visited during the New Year period, and relatives and friends who did not visit during the New Year period on the purchase of policy-based planting agricultural insurance by the interviewed farmers. In the second step, according to the propensity score, different matching algorithms were used to match the samples to control the selection bias of the samples. The third step, on the basis of the matching samples, compared the average difference between the treatment group and the control group, and then obtained the causal coefficient, that is, the ATT value, which is defined as:(3)$$\mathrm{ATT}=\;E\left\{E\left[Y_{1i}\left|D_{i}=1,\;P\left(X_{i}\right)\right.\right]-E\left[Y_{0i}\left|D_{i}=0\right.,\;P\left(X_{i}\right)\right]\left|D_{i}=1\right.\right\}$$

In the formula, $D_{i}$ is a binary variable used to report whether individual i belongs to the control group; $P\left(X_{i}\right)$ represents the propensity score; and $Y_{1i}$ and $Y_{0i}$ represent the estimated results of different groups, respectively.

## 3. Results and Discussion

### 3.1. Results of the Binary Logistic Model

The purpose of this study was to reveal the influence of the peer effect on farmers’ purchases of policy-based planting agricultural insurance. As shown in Table 2, Model 1 reports the results in which only the core explanatory variables enter the estimation; Model 2 reports the results in which the core explanatory variables, control variables, and dummy variables for the controlled area enter the estimation; and the last column reports the marginal effects. It can be seen from the table that, even if the control variables and the regional dummy variables exist, the conformity tendency of relatives and friends, relatives and friends who visited during the New Year period, and relatives and friends who did not visit during the New Year period also have a significant impact on farmers’ decisions to purchase policy-based planting agricultural insurance, and the parameters all increase, which is in line with our expectations. The main focus of this paper is the regression results of Model 2.

The regression results of Model 2 reveal that the purchase of policy-based planting agricultural insurance by relatives and friends has a significant positive impact on the purchase of policy-based planting agricultural insurance by farmers. Combined with the results of the marginal effect, it was found that, for each 1% increase in the value of the variable consistent with relatives and friends, the probability of farmers purchasing policy-based planting agricultural insurance increased by 50.160%. The possible reason for this result is that today’s society is an information society, and the speed at which information is replaced is changing daily. Due to information media and individual differences, information available to farmers is limited and prone to deviation. Regarding this type of policy insurance, in particular, farmers often lack a correct and comprehensive understanding due to various reasons, such as the publicity methods and the strengths of policies in various locations in China, and farmers’ previous experience and enthusiasm for active learning related to insurance. In rural areas where education levels are generally low and information is blocked, farmers may experience information bias and hesitant judgments. At this time, farmers’ compliance with relatives and friends may lead the farmers to follow or learn from relatives and friends.

As can be seen from Model 2, the consistency of relatives and friends who visit during the New Year period has a significant positive impact on farmers’ purchase of policy-based planting agricultural insurance. Combining the results of the marginal effect, it can be seen that, for each 1% increase in the conformity of farmers to relatives and friends who visited during the New Year period, the probability that the family is willing to adopt insurance increases by 38.414%. One possible explanation for this finding is that China has traditionally been a society of acquaintances. In rural areas, “people around” who have blood ties or friendships with farmers often communicate and interact with farmers; that is, the decisions, behaviors, and choices of relatives and friends who visit during the New Year period are often the first choice for farmers. This often leads to herd tendencies and behaviors.

It can be seen from Model 2 that the consistency of relatives and friends who did not visit during the New Year period has a significant positive impact on farmers’ purchase of policy-based planting agricultural insurance. Combining the results of the marginal effect, it can be seen that, for each 1% increase in the conformity of farmers to relatives and friends who did not visit during the New Year period, the probability that the family is willing to adopt insurance will increase by 28.160%. A possible explanation for this finding is that, because China is traditionally a society of acquaintances, even if farmers have frequent exchanges with relatives and friends who do not visit during the New Year period, they will still receive information through other channels (such as village cadres or insurance company formulas). A certain degree of news, in this case, the decisions, behaviors, and choices of those relatives and friends who do not visit during the New Year period, will often arouse the interest of farmers, thereby further deepening their understanding of their decisions, behaviors, and choices. This leads to the tendency and behavior of conformity of farmers.

It is noted that, although the three core explanatory variables of relatives and friends, relatives and friends who visited during the New Year period, and relatives and friends who did not visit during the New Year period all have a significant positive impact on farmers’ purchases of policy-based planting agricultural insurance, the degree of impact is significantly different. Relatives and friends have the greatest impact on the purchase behavior of farmers (marginal effect 50.160%), followed by relatives and friends who visited during the New Year period (marginal effect 38.414%), and those who did not visit during the New Year period (marginal effect 28.160%). The decreasing value of the marginal effect is also consistent with the analysis of strong and weak relationships undertaken earlier in this study. The possible explanation for this finding is that the relationships among relatives and friends also vary in terms of distance. Compared with relatives and friends who do not visit during the New Year period, those who represent strong relationships make contact during the New Year period. Relatives and friends usually have a higher frequency of interaction and similar attitudes to farmers, so their decisions, behaviors, and choices are often the first choice for farmers.

In conclusion, the results of Models 1 and 2 both confirm the main expectation of this study; that is, the purchase of policy-based planting agricultural insurance by relatives and friends, relatives and friends who visited during the New Year period, and relatives and friends who did not visit during the New Year period have a significant impact on farmers’ purchases of policy-based planting agricultural insurance. In reality, however, there are many similarities and differences between relatives and friends who meet the understanding of farmers, relatives and friends who visit during the New Year period, and relatives and friends who do not visit during the New Year period. This means that an individual’s tendency to conform is probably the result of self-selection. Therefore, the direct regression results of the binary logistic model may have selection bias. It can also be seen from the regression results of Model 2 that many variables affect farmers’ conformity tendencies, such as education level and land management area. In this context, we chose to build a counterfactual framework (PSM model) for bias correction and robustness testing, thereby avoiding selection bias.

### 3.2. Correction Results of the PSM Model

In order to further avoid selection bias, address the potential endogeneity problem of key variables, and ensure the robustness of the regression results, we established a PSM model using three matching algorithms: one-to-one nearest neighbor matching, one-to-four nearest neighbor matching, and kernel matching. These were used to further estimate the peer effect on farmers’ decisions to purchase policy-based planting agricultural insurance. The distribution of propensity scores and areas of common support are specifically portrayed in Figure 2. The data in the figure clearly reveal the importance of appropriate matching and imposing common support conditions to avoid bad matching. It can also be seen from the figure that the three samples under the matching algorithm are matched, and all differences are significantly reduced. Each of the three matching algorithms was repeated 500 times, and the specific regression results are shown in Table 3. The results of ATT show that, after avoiding the observable systematic differences between samples, the peer effect still has a significant impact on farmers’ decisions to purchase policy-based planting agricultural insurance. This is also in good agreement with the regression results of the binary logistic model.

First, regarding the peer effect of friends and relatives, the ATT values obtained under the three matching algorithms are all positive. It can also be judged from the values that this is the most positive of the three core explanatory variables, all of which have statistics at the 1% level. Among the three matching algorithms, the ATT value obtained by kernel matching, of 0.564, is the largest, followed by 0.552 for the one-to-one nearest neighbor matching method and 0.532 for the one-to-four nearest neighbor matching method. Although the ATT values obtained by different matching methods are slightly different, the results uniformly show that, under the influence of relatives and friends purchasing policy-based planting agricultural insurance, the possibility of farmers purchasing policy-based planting agricultural insurance is increased.

Secondly, for the peer effect of relatives and friends who visited during the New Year period, the ATT values obtained under the three matching algorithms are all positive and statistically significant at the 1% level. The ATT value obtained based on the kernel matching algorithm, of 0.502, is the largest, followed by 0.488 for the one-to-four nearest neighbor matching method and 0.460 for the one-to-one nearest neighbor matching method. Although the significance and value of the ATT values obtained by different matching methods are slightly different, the results show that the purchase by relatives and friends who visited during the New Year period of policy-based planting agricultural insurance tends to encourage farmers to purchase policy-based planting agricultural insurance.

Regarding the peer effect of relatives and friends who did not visit during the New Year period, the ATT values obtained under the three matching algorithms are all positive and statistically significant at the 1% level. The ATT value based on the one-to-four nearest neighbor matching method, of 0.332, is the largest, followed by 0.329 for the one-to-one nearest neighbor matching method and 0.323 for the kernel matching algorithm. Although the significance and value of the ATT values obtained by different matching methods are slightly different, the results show that the purchase by relatives and friends who did not visit during the New Year period of policy-based planting agricultural insurance tends to encourage farmers to purchase policy-based planting agricultural insurance.

Finally, it can be seen from the table that, in each matching algorithm, the ATT value of relatives and friends is greater than the ATT value of relatives and friends who visited during the New Year period, and the ATT value of relatives and friends who visited during the New Year period is greater than the ATT value of relatives and friends who did not visit during the New Year period. This finding is also in good agreement with the regression results of the binary logistic model. It shows that relatives and friends have the greatest influence on the peer effect of farmers, followed by relatives and friends who visited during the New Year period, and then relatives and friends who did not visit during the New Year period.

### 3.3. Mechanism Analysis

The underlying mechanism of the impact of the peer effect on the insurance purchase behavior of farmers remains to be further revealed. China is a society of acquaintances, and the social network relationship of farmers can represent the communication, contact, and interaction between individuals and those around them [50,51,52,53,54,55,56]. Therefore, this study used two entries that reflect the interaction between farmers and relatives and friends: “when farming is busy, you will often go to relatives and friends to help” and “if relatives and friends (not in the same village) have important matters, they will often consult with you”. These entries reflect the social network relationship of farmers. Secondly, because China traditionally comprises a society of acquaintances, trust often has a low external interaction. People will clearly divide themselves from outsiders based on the scope of trust and on the basis of mutual understanding. This kind of trust is known as relational trust, which manifests itself in decreasing levels of trust in relatives, friends, and strangers. After trust breaks through the scope of the blood relationship, it is difficult to expand each step outward [29,57]. Generally speaking, the higher the credibility of the local government, the stronger the appeal, and the more active the residents’ participation in public policies [58]. Therefore, this study used two terms—”trust in strangers” and “trust in local government”—to analyze the mechanism of farmers’ trust.

The two mechanisms analyzed in this study were: peer effect→social network→purchase policy-based planting agricultural insurance; and peer effect→trust→purchase policy-based planting agricultural insurance.

**Hypothesis** **5** **(H5).**
*The stronger the social network of farmers, the more likely they will be to buy policy-based planting agricultural insurance.*


**Hypothesis** **6** **(H6).**
*The greater the farmers’ trust in the population, as represented by the three core variables, the more likely they will be to purchase policy-based planting agricultural insurance.*


In order to verify the above two mechanisms, this study adopted the mediation effect analysis method [56] to conduct the empirical research. The model is as follows:(4)$$\;Y=cX_{i}+\varepsilon_{1i}\;$$
(5)$$\;M_{i}=\alpha X_{i}+\varepsilon_{2i}\;$$
(6)$$\;Y=c^{\prime }X_{i}+\beta M_{i}+\varepsilon_{3}\;$$
where Y represents the farmer’s purchase of policy-based planting agricultural insurance, $X_{i}$ represents the core explanatory variable reflecting the peer effect, and $M_{i}$ is the mediating variable in the mechanism, representing the social network relationship of the farmer and the trust of the farmer, respectively. All models used the stepwise regression command. The results are shown in Table 4, which reports the marginal effect results for both mechanisms.

Regarding the regression results of Mechanism 1, the three core explanatory variables reflecting the peer effect and the purchase of policy-based planting agricultural insurance by farmers are significantly positive at the 1% level, whereas the influence of relatives and friends on the two mediating variables in the mechanism is not significant. In the whole mechanism, only the two paths of relatives and friends who did not visit during the New Year period→social network→purchase policy-based planting agricultural insurance are significant. When two mediator variables representing social networks were added to the model, the marginal influence coefficient of the peer effect on farmers’ decisions to purchase policy-based planting agricultural insurance behavior decreased from 0.322 to 0.023 and 0.028, but the obtained marginal influence coefficient was significantly positive at the 1% level. Therefore, H3 was verified, but the results prove that H5 does not hold. The explanation for this result may be that, for those relatives and friends who did not visit during the New Year period, their communication, contact, and interaction with farmers are not frequent or close. Therefore, if the impact of the peer effect is simply measured through social networks, the impact of relatives and friends who did not visit during the New Year period will be reduced.

Regarding the regression results of Mechanism 2, the three core explanatory variables reflecting the peer effect and farmers’ purchase of policy-based planting agricultural insurance are significantly positive at the 1% level, whereas the two core explanatory variables of relatives and friends, and relatives and friends who visited during the New Year period, are positive. The effect of the mediating variable “trust in strangers” in Mechanism 2 was not significant. In the whole mechanism, three paths were significant: relatives and friends who did not visit during the New Year period→trust in strangers→purchase policy-based planting agricultural insurance; relatives and friends who did not visit during the New Year period→trust in the government→purchase policy-based planting agricultural insurance; and relatives and friends→trust in the government→purchase policy-based planting agricultural insurance. In these paths, when the two intermediary variables representing trust were added to the model, the marginal influence coefficient of relatives and friends on farmers’ decisions to purchase policy-based planting agricultural insurance dropped from 0.541 to 0.035, but the marginal influence coefficient obtained was significantly positive at the 1% level, thereby verifying H1. The marginal influence coefficient of relatives and friends who did not visit during the New Year period on farmers’ purchases of policy-based planting agricultural insurance dropped from 0.322 to −0.037 and 0.039. This also verifies H6. The explanation for this result may be that the government’s credibility and appeal affect the decision-making behavior of the overall group of relatives and friends, thus affecting the decision-making behavior of farmers. However, it can be seen from the decline in the marginal influence coefficient that the local government’s credibility and appeal are not strong. Regarding relatives and friends who do not visit during the New Year period, their communication, contact, and interaction with farmers are not frequent or close. Therefore, if the impact of the peer effect is simply measured by trust, the impact of relatives and friends who do not visit during the New Year period will be reduced or even be negative.

The results of the above two mechanisms show that, through the social network and trust of farmers, the impact of the peer effect on farmers’ purchases of policy-based planting agricultural insurance is reduced. Therefore, the most effective means to more actively encourage farmers to purchase policy-based planting agricultural insurance is to directly publicize to households or to promote dissemination through popular farmers having strong appeal in the village.

### 3.4. Heterogeneity Analysis

Analysis presented in the previous section verified that the peer effect can significantly promote the purchase of policy-based planting agricultural insurance by farmers and revealed its internal mechanism. However, the above results represent the average effect of the whole sample, and it is necessary to further discuss whether there are differences between different farmer groups. Theoretically, farmers having larger land scales are more worried about the impacts of meteorological disasters, pests, and natural disasters on crop yields. Farmers having higher education levels have better awareness and acceptance of insurance, and tend to be more inclined to purchase policy-based planting agricultural insurance [12,39,59,60,61,62,63]. Based on this, further heterogeneity analysis was carried out at the household and individual levels according to the size of the land under management and the education level of the respondents (Table 5).

At the household level, the three core explanatory variables that reflect the peer effect have a significant impact on the purchase of policy-based planting agricultural insurance by the farmers whose land scale is below the average level of 5.679 mu, and those above the average level, at the level of 1%. This result validates H1, H2, and H3. Comparing the results, it can be seen that the larger the farmer’s land scale, the greater the impact of the peer effect, which verifies H4. The result also shows that, the larger the scale of farmland, the more disasters suffered by the land will be considered in advance, thereby avoiding risk and providing greater protection of agricultural income.

At the individual level, the three core explanatory variables reflecting the peer effect on the purchase of policy-based planting agricultural insurance by farmers who received 9 years of compulsory education, and farmers who received more than 9 years of education, are all significant at the 1% level. This result validates H1, H2, and H3. Comparing the results, it can be seen that farmers who received higher education will be more affected by the peer effect, and thus more willing to buy policy-based planting agricultural insurance, which verifies H4. This also shows that the higher the education level of the farmers, the greater their degree of accepting new things, the greater their degree of receiving and understanding news, and the greater their degree of risk aversion.

## 4. Conclusions

Based on the survey data of 540 farmer households in Sichuan Province, using a binary logistic regression model combined with PSM and other methods to systematically analyze the peer effect on the purchase by farmer households of policy-based planting agricultural insurance, and its mechanism, the main conclusions are as follows: (1) Farmers’ decisions to purchase policy-based planting agricultural insurance are influenced by their relatives and friends, and this influence has a “circle effect”; that is, the decisions to purchase policy-based planting agricultural insurance by relatives and friends who visit during the New Year period have a greater influence than those of relatives and friends who do not visit during the New Year period. (2) The results of the mechanism analysis show that, through the mediating variables of social networks and trust, the influence of the peer effect on the purchase of policy-based planting agricultural insurance by farmers is weakened. (3) Heterogeneity analysis shows that farmers having larger land scales and higher education levels are more affected by the peer effect and are more inclined to purchase policy-based planting agricultural insurance.

## 5. Policy Recommendations

Based on the previous literature, this study proposes the following policy recommendations. (1) Regarding the state, the promotion of policy-based planting agricultural insurance is inseparable from the policy support of the state. The state should improve the use efficiency of financial funds, continue to increase financial subsidies for insurance premiums, replace policy-based planting agricultural insurance that is not suitable for local governments, and continuously expand the scope of policy-based planting agricultural insurance. (2) Insurance institutions providing policy-based planting agricultural insurance should conduct surveys and determine damage to severely damaged plots in a timely manner, and settle claims in a reasonable manner. The development of policy-based planting agricultural insurance products should be strengthened, and policy-based planting agricultural insurance services should be strengthened and improved. (3) It is necessary for the local government to strengthen the contact with the higher-level government to be familiar with the latest policy trends and most recent information. (4) Farmers should actively participate in agricultural technology training and policy publicity meetings organized by local organizations, so as to promote themselves from an attitude of “I don’t want to protect”, to one of “I want to protect”, and to actively learn the approaches of advanced farmers around them.

Finally, although this research was mainly aimed at China, which has Confucian culture as its background, it can be used to a further research, and, ultimately, can be applied to other regions deeply influenced by Confucian culture, such as Singapore. For regions that are not influenced by Confucian culture, such as Africa, a similar design idea can also be provided. For example, the strong and weak relationships of individuals can be represented by certain behaviors in line with the local social and cultural backgrounds, to study the influences on the purchasing behavior of farmers and the impact of the circle effect.

## Figures and Tables

**Figure 1 ijerph-19-07411-f001:**
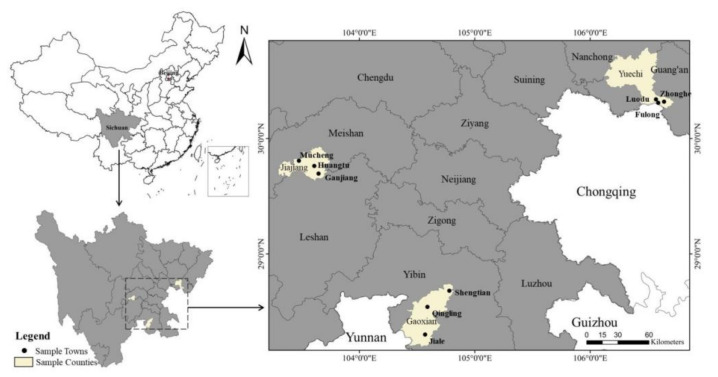
Distribution map of the sample area.

**Figure 2 ijerph-19-07411-f002:**
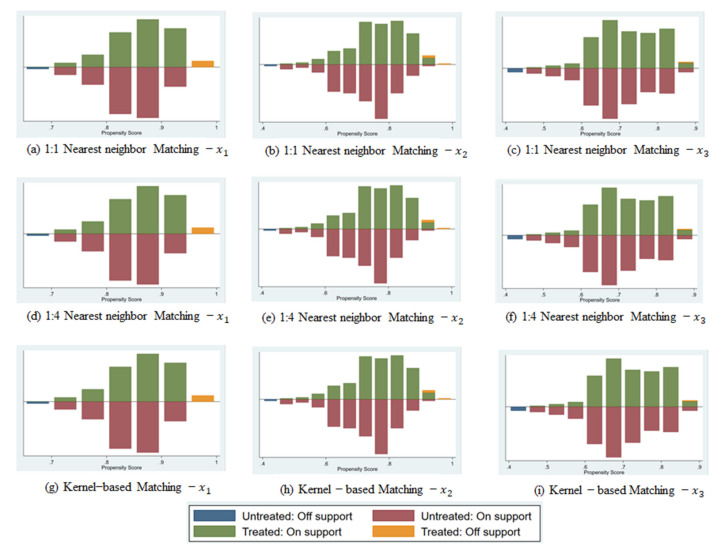
Influence of the cohort effect after bias correction.

**Table 1 ijerph-19-07411-t001:** Variable definitions and descriptive statistics.

Variable	Variable Measure	Average	Standard Deviation
Y	Do you have policy-based planting agricultural insurance? ^a^	0.620	0.486
$x_{1}$	Have your relatives and friends purchased this type of insurance? ^a^	0.863	0.344
$x_{2}$	Have your relatives and friends who visited during New the Year period purchased this type of insurance? ^a,d^	0.761	0.427
$x_{3}$	Have your relatives and friends who did not visit during the New Year period purchased this type of insurance? ^a,d^	0.713	0.453
gender	Your gender (0 = male, 1 = female)	0.404	0.491
age	Your age (years)	58.480	11.840
education	Your education level (years)	6.552	3.443
Farming time	How long have you been farming? (year)	38.220	14.74
Family-owned land area	Land area under operation in 2020 (mu) ^c^	5.679	20.532
income	Annual household income in 2020 (yuan)	92,992.728	168,723.286
Disaster perception	Will crops suffer/reduce production due to disasters/weather in 2020? ^a^	0.702	0.458
Environmental awareness	Are you very concerned about the severe impact of climate change on agricultural production? ^a^	4.200	1.032
Willingness for rural development	Your willingness to continue farming ^b^	3.644	1.393
Risk aversion	Two kinds of investments are available to choose from: if you choose the first, you have a 100% chance of receiving CNY 5000; if you choose the second, you have a 50% chance of receiving CNY 10,000, and a 50% chance of receiving nothing. Which would you choose? (0 = first type,1 = second)	0.096	0.295
county_1	county_1 ^a^	0.333	0.472
county_2	county_2 ^a^	0.333	0.472
county_3	county_3 ^a^	0.333	0.472

Note: ^a^ score (0 = no, 1 = yes); ^b^ Likert 5-point scale, where 1 means strongly disagree and 5 means strongly agree; ^c^ 1 mu = 0.067 hectares; ^d^ The Spring Festival is the grandest traditional festival in China, and New Year’s greetings are typical behaviors that best reflect the strength of social exchanges and human relationships.

**Table 2 ijerph-19-07411-t002:** Results of the binary logistic model.

Variable	Model 1			Model 2			Marginal Effect
	$x_{1}$	$x_{2}$	$x_{3}$	$x_{1}$	$x_{2}$	$x_{3}$	
$x_{1}$	2.966 *** (0.388)			3.091 *** (0.396)			0.502 *** (0.053)
$x_{2}$		2.425 *** (0.245)			2.468 *** (0.270)		0.384 *** (0.027)
$x_{3}$			1.537 *** (0.203)			1.604 *** (0.223)	0.282 *** (0.031)
gender				0.008 (0.242)	−0.095 (0.248)	−0.068 (0.233)	
age				−0.022 (0.017)	−0.016 (0.017)	−0.030 * (0.017)	
education				0.079 ** (0.037)	0.080 ** (0.037)	0.063 * (0.035)	
Farming time				0.023 * (0.013)	0.017 (0.013)	0.032 ** (0.013)	
Family-owned land area				0.399 ** (0.172)	0.392 ** (0.170)	0.468 *** (0.164)	
income				−0.026 (0.099)	0.015 (0.097)	0.038 (0.095)	
Disaster perception				0.635 *** (0.241)	0.455 * (0.255)	0.727 *** (0.233)	
Environmental awareness				0.179 (0.109)	0.212 * (0.110)	0.109 (0.109)	
Willingness for rural development				0.112 (0.079)	0.144 * (0.082)	0.074 (0.079)	
Risk aversion				0.780 (0.483)	0.698* (0.414)	0.856 ** (0.434)	
county_1				−0.511 * (0.273)	−0.406 (0.275)	−0.445 * (0.256)	
county_2				1.251 *** (0.309)	1.290 *** (0.312)	1.239 *** (0.301)	
Log likelihood	−309.330	−297.575	−327.960	−265.758	−258.341	−283.601	
Prob > chi^2^	0.0000	0.0000	0.0000	0.000	0.000	0.000	
Pseudo R^2^	0.1371	0.1699	0.0852	0.259	0.279	0.209	
N	540	540	540	540.000	540.000	540.000	

Note: * means *p* < 0.1, ** means *p* < 0.05 and *** means *p* < 0.001.

**Table 3 ijerph-19-07411-t003:** Average processing effect of different matching algorithms based on PSM.

Matching Algorithms	Influencing Factors	ATT	Std. Err.	Treated	Controls
Nearest neighbor matching (1:1)	$x_{1}$	0.552 *** (8.92)	0.062	0.693	0.141
	$x_{2}$	0.460 *** (7.86)	0.059	0.745	0.285
	$x_{3}$	0.329 *** (5.38)	0.061	0. 723	0.394
Nearest neighbor matching (1:4)	$x_{1}$	0.532 *** (10.79)	0.049	0.693	0.161
	$x_{2}$	0.488 *** (9.92)	0.049	0.745	0.257
	$x_{3}$	0.332 *** (6.59)	0.050	0.723	0.392
Kernel-based matching (bandwidth 0.06)	$x_{1}$	0.564 *** (12.73)	0.044	0.693	0.129
	$x_{2}$	0.502 *** (11.17)	0.045	0.745	0.243
	$x_{3}$	0.323 *** (6.83)	0.047	0.723	0.400

Note: *** means *p* < 0.001.

**Table 4 ijerph-19-07411-t004:** Mechanism analysis.

Variable	Mechanism 1: Peer Effect→Social Network→Purchase Policy-Based Planting Agricultural Insurance						Mechanism 2: Peer Effect→Trust→Buy Policy-Based Planting Agricultural Insurance					
	Model 3			Model 4			Model 5			Model 6		
	X→Y	X→M	X→M→Y	X→Y	X→M	X→M→Y	X→Y	X→M	X→M→Y	X→Y	X→M	X→M→Y
$x_{1}$	0.541 ***	−0.251	0.024 *	0.541 ***	0.232	0.027 *	0.541 ***	−0.024	−0.019	0.541 ***	0.212 *	0.035 *
	(0.052)	(0.184)	(0.012)	(0.052)	(0.159)	(0.014)	(0.052)	(0.111)	(0.021)	(0.052)	(0.109)	(0.021)
$x_{2}$	0.483 ***	0.355 **	0.017	0.483 ***	0.261 **	0.023	0.483 ***	−0.127	−0.007	0.483 ***	0.100	0.041 **
	(0.042)	(0.149)	(0.012)	(0.042)	(0.129)	(0.014)	(0.042)	0.090	0.020	(0.042)	(0.089)	(0.020)
$x_{2}$	0.322 ***	0.279 **	0.023 *	0.322 ***	0.218 *	0.028 *	0.322 ***	0.187 **	−0.037 *	0.322 ***	0.161 *	0.039 *
	(0.041)	(0.140)	(0.013)	(0.041)	(0.121)	(0.015)	(0.042)	(0.084)	(0.021)	(0.041)	(0.084)	(0.022)
Control	Yes	Yes	Yes	Yes	Yes	Yes	Yes	Yes	Yes	Yes	Yes	Yes
County	Yes	Yes	Yes	Yes	Yes	Yes	Yes	Yes	Yes	Yes	Yes	Yes

Note: * means *p* < 0.1, ** means *p* < 0.05 and *** means *p* < 0.001.

**Table 5 ijerph-19-07411-t005:** Heterogeneity analysis.

Variable	The Size of the Family-Run Land						Respondent’s Educational Level					
	The Average Household Owns 5.679 mu and Below			The Average Household Occupies More than 5.679 mu			9 Years and Below			Over Nine Years		
$x_{1}$	3.038 ***			3.928 ***			3.054 ***			4.916 ***		
	(0.438)			(0.964)			(0.419)			(1.443)		
$x_{2}$		2.348 ***			3.366 ***			2.361 ***			3.657 **	
		(0.308)			(0.662)			(0.289)			(1.712)	
$x_{3}$			1.590 ***			1.869 ***			1.636 ***			3.097 *
			(0.265)			(0.476)			(0.237)			(1.752)
Control	Yes	Yes	Yes	Yes	Yes	Yes	Yes	Yes	Yes	Yes	Yes	Yes
County	Yes	Yes	Yes	Yes	Yes	Yes	Yes	Yes	Yes	Yes	Yes	Yes
Wald χ^2^	109.62 ***	97.34 ***	86.31 ***	45.01 **	42.67 ***	35.55 ***	106.84 ***	103.24 ***	86.95 ***	31.25 ***	28.37 ***	15.68
Observation	414	414	414	126	126	126	478	478	478	62	62	62

Note: * means *p* < 0.1, ** means *p* < 0.05 and *** means *p* < 0.001.

## Data Availability

Not applicable.

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
