# Peer review of "Peer Effects on Farmers’ Purchases of Policy-Based Planting Farming Agricultural Insurance: Evidence from Sichuan Province, China"

_ijerph, 2022, doi:10.3390/ijerph19127411_

Round 1
Reviewer 1 Report
The manuscript is improved in terms of presentation but I still have not seen the presentation of what kind of losses and percentage are covered. Are there any deductibles and what percentage of total premium is paid by the government in terms of subsidies?
Your results show that elasticity variables are too high (i.e. anywhere from 28% to more than 50%). You need to explain what is driving these results of having too sensitive values, otherwise readers are going to question the accuracy or the way data were collected.
In general, there are too many grammar errors in the manuscript and I cannot go through line by line to correct them. It needs heavy proofreading before acceptance.
You are using too frequently a “policy-based planting” wording. Explain at the beginning of the manuscript why do you need to use that wording and then mention that from thereafter it will be assumed that or use it sparingly.
Also, in lines 70-74 you are basically repeating the same sentence twice.
Line 322 – correct typo ‘com-pares’, same in line 394 ‘rela-tives’ and line 447 'alt-hough'. There are many instances like this in the manuscript.
Author Response
Dear Reviewer:
Thank you for your valuable suggestions. In response to your valuable suggestions, I have made the following changes to the manuscript:
- In response to your questions about grammar errors in the manuscript, sentence repetition problems in 70-74 and word correction problems, the author once again proofread the full text and corrected them one by one.
- In response to your questions about "the presentation of what kind of losses and percentage", the original manuscript mentioned the area affected by floods and droughts in China's planting industry in 2019. Regarding your mention of "Are there any deductibles and what percentage of total premium is paid by the government in terms of subsidies", the author reintroduced the description in the original manuscript: at present, for policy-based crop farming agricultural insurance, the central financial subsidy is 40%, the provincial financial subsidy is 25%, the city and county financial subsidy is 15%, and the grower bears 20%. At the same time, the author also added the specific content about the premium subsidy and the insurance amount compensation in the introduction: the scope of compensation for policy-based agricultural insurance includes: the amount of planting insurance is based on the direct physical and chemical costs incurred during the growth period of the insurance subject(including costs of seeds, fertilizers, pesticides, irrigation, machine farming and mulch film, etc.) ; the insurance amount for aquaculture is the insured individual Physiological value (including purchase cost and feeding cost). Specifically, at present, the specific compensation standard of China's policy-based crop farming insurance is: the starting point for claims is 30%, that is, when the loss rate of the insured crops due to natural disasters reaches 30% (including 30%) to 70%, according to The crop growth period is divided into the insurance amount and the loss rate to calculate the compensation. The specific calculation formula for the compensation is: compensation amount = insurance amount in each growth period loss rate damaged area. When the loss rate reaches more than 70%, the full amount of insurance will be paid according to the crop growth period. Taking rice cultivation as an example, the insurer is not responsible for compensation due to natural disasters within the scope of insurance coverage but the loss rate is less than 30%; if the actual loss rate is above 30% (including 30%), the compensation will be paid proportionally, 70% (including 70%) or more full compensation. When the area of the insured rice plot of each insured is smaller than the actual planting area, the compensation shall be calculated according to the proportion of the insured area to the actual planting area.
- In response to your question about elasticity variables are too high. The author has added relevant explanations, explanations and speculations in the manuscript analysis section. For example: Among them, relatives and friends have the greatest impact on the purchase behavior of farmers(marginal effect 50.160%), followed by relatives and friends who have New Year visits(marginal effect 38.414%), and those who do not have the least impact(marginal effect 28.160%). The decreasing value of marginal effect is also consistent with the analysis of strong relationship and weak relationship earlier in this study. The possible explanation for this finding is that relatives and friends also have distance and distance. Compared with relatives and friends who have no New Year's greetings, those who represent strong relationships have New Year's contacts. Relatives and friends usually have a higher frequency of interaction and similar attitudes with farmers, so their decisions, behaviors and choices are often the first choice for farmers.
- In response to your questions about frequently using a "policy-based planting” wording, the first is to prevent confusion of concepts, because policy-based crop farming agricultural insurance is only one of the types of policy-based agricultural insurance in China. At present, China's policy-based agricultural insurance includes a total of 16 varieties such as planting, aquaculture, and forestry. Secondly, because many concepts about insurance are involved or juxtaposed, the manuscript clearly informs the readers that the policy-based agricultural insurance is the only object of this research. Finally, because the introduction part of the review also involves policy-based agricultural insurance and other types of insurance, in order to clearly distinguish the concept and use rigorous words. Therefore, the abbreviation of "policy-based crop farming agricultural insurance" was not considered when it first appeared.
Thank you again for your valuable advice and wish you a happy life.
Yours faithfully
Reviewer 2 Report
Dear Authors
Thank you for including my suggestions and comments in the revised work.
Reviewer
Author Response
Dear Reviewer:
Thank you again for your valuable advice and wish you a happy life.
Yours faithfully
This manuscript is a resubmission of an earlier submission. The following is a list of the peer review reports and author responses from that submission.
Round 1
Reviewer 1 Report
The manuscript needs some revisions.
Line specific comments:
Line 40 - rephrase the sentence
Line 50 - the authors are mentioning "premium subsidies will reach x amount in 2021", but it is already 2022. Did it happen? If so, provide the source.
Lines 63-66 - rephrase the sentence.
Line 68: I do not see how it is anomaly because if the program is relatively new and is being tested and more than 55% of crops are not covered, it may take time before the farmers realize and strong adoption takes place.
Lines 138 - 145: the sentence needs heavy rephrasing.
Lines 168 - 172: hard to follow this sentence.
Line 220: split "and"
Line 247: The sentence is off. What do you mean by risk averse decision-makers may refuse to purchase insurance? We expect, more risk averse a farmer is, more likely he/she will purchase an insurance. Teh sentence needs more explanations.
General comments:
The authors did not explain what kind of farmers have been analyzed. What kind of crop farmers are they? Are they all grouped together? Are there any livestock farmers?
The authors did not provide any details regarding what kind of insurance product is it. Is it yield insurance? Is it revenue insurance? Are there any coverage levels available? What portion of total premium rates are covered by the government? Are the indemnities limited to certain conditions?
Reviewer 2 Report
Dear Authors
In my opinion, this scientific article is a methodological work related to the management of agricultural insurance. The authors of the study did not convince me that the research undertaken in this way would explain the relationship between changes in the environment, manifested by losses in agricultural production, and, for example, the problem of hunger in a given region or the economic decline of farms not using agricultural insurance. The conclusions presented in the article are mainly used in agricultural management in China.
The holiday period, e.g. in European countries, is divided into several periods. Hence, some of the valuable proposals for improving management will not be able to be applied in other regions of the world.
Therefore, I am asking the authors of the work to rewrite the manuscript in the direction of indicating, first of all, the importance of the conducted research for the development of the methodology of evaluating the purchase of agricultural insurance guaranteeing sustainable development of rural areas. On the basis of many other scientific studies, please show the importance of the peer effect on the purchase of products and services. It would be valuable to present the effects of climate change in the area and the trends in purchasing insurance by farmers. Please comment on the results of the research in the context of their importance, for example, for Africa.
Please introduce to the methodology the assumptions of Mechanism analysis and Heterogeneity Analysis, which were discussed in the study results chapter.
Yours faithfully
Reviewer